# New Insights in the Detection and Management of Anthracnose Diseases in Strawberries

**DOI:** 10.3390/plants12213704

**Published:** 2023-10-27

**Authors:** Baker D. Aljawasim, Jayesh B. Samtani, Mahfuzur Rahman

**Affiliations:** 1Hampton Roads Agricultural Research and Extension Center, School of Plant and Environmental Sciences, Virginia Polytechnic Institute and State University, 1444 Diamond Springs Road, Virginia Beach, VA 23455, USA; bdal222@vt.edu; 2Department of Plant Protection, College of Agriculture, Al-Muthanna University, Samawah 66001, Iraq; 3Extension Service, Davis College of Agriculture, West Virginia University, 1194 Evansdale Drive, Morgantown, WV 26506, USA

**Keywords:** anthracnose fruit rot, *Colletotrichum acutatum*, diagnostics, crop protection, sustainable management techniques, strawberry

## Abstract

Anthracnose diseases, caused by *Colletotrichum* spp., are considered to be among the most destructive diseases that have a significant impact on the global production of strawberries. These diseases alone can cause up to 70% yield loss in North America. *Colletotrichum* spp. causes several disease symptoms on strawberry plants, including root, fruit, and crown rot, lesions on petioles and runners, and irregular black spots on the leaf. In many cases, a lower level of infection on foliage remains non-symptomatic (quiescent), posing a challenge to growers as these plants can be a significant source of inoculum for the fruiting field. Reliable detection methods for quiescent infection should play an important role in preventing infected plants’ entry into the production system or guiding growers to take appropriate preventative measures to control the disease. This review aims to examine both conventional and emerging approaches for detecting anthracnose disease in the early stages of the disease cycle, with a focus on newly emerging techniques such as remote sensing, especially using unmanned aerial vehicles (UAV) equipped with multispectral sensors. Further, we focused on the *acutatum* species complex, including the latest taxonomy, the complex life cycle, and the epidemiology of the disease. Additionally, we highlighted the extensive spectrum of management techniques against anthracnose diseases on strawberries and their challenges, with a special focus on new emerging sustainable management techniques that can be utilized in organic strawberry systems.

## 1. Introduction

Strawberry (*Fragaria ananassa* Duchesne), a major small fruit across the world, has an attractive flavor and taste and a high content of essential nutrients that benefit human health [1]. The Food and Agriculture Organization of the United Nations (FAO) reports that strawberries were planted on 389,665 ha and produced about 9,175,384.43 metric tons globally in 2021 [2]. The United States of America is among the countries with the highest strawberry production and plants approximately 23,500 ha of strawberries, with a value of US $2.4 billion [3]. In addition, the south-Atlantic region of the United States produces 947 ha of strawberries, with an average yield of 15,668 kg/ha and a total farm gate value of $47,158,000 [4]. Strawberry consumption in the United States has grown over the last two decades, from 0.9 kg per capita in 1980 to 3.6 kg in 2013 [5]. However, diseases and pests significantly lower the quality and yield of strawberry fruit, incurring significant financial losses for growers.

The mid-Atlantic region of the United States, including the Commonwealth of Virginia, ranks third in the production of fresh market strawberries after California and Florida, and most growers are using the annual hill plasticulture (AHP) production system, likely due to the disease risk from the buildup of inoculum in a perennial system [4]. Field preparation begins with debris removal, disking, and tilling the soil, followed by bedding, plastic covering, drip tape installation, and overhead watering in September. In most cases, the raised soil beds are fumigated to control nematodes, fungi, and weeds. However, due to new regulations, fumigation with synthetic chemicals is becoming difficult for small growers and farms close to public places. Then, the transplants are planted and overhead watered for establishment between mid-September and early October, when temperatures couldreach above 30 °C [6,7]. Strawberry fruits are harvested two to three times each week on average when the berries are fully ripe, from mid-April until late June. The strawberry plant can be infected by different organisms, including many arthropods, nematodes, fungi, bacteria, viruses, and other pests [8]. Further, the strawberry plant is highly susceptible to a large variety of soilborne pathogens, including the genera *Verticillium*, *Phytophthora*, and *Colletotrichum*, which are considered the most damaging pathogens on this crop in the United States strawberry production system [8,9]. In general, anthracnose diseases are caused by fungal pathogens belonging to multiple species under the acutatum species complex and the gloeosporioides species complex of the genus *Colletotrichum*. All of these pathogens can cause infection on any part of a strawberry; however, the acutatum species complex tends to be more destructive as fruit rot pathogens, and the gloeosporioides species complex is more damaging as crown rot pathogens. Considering the overall importance and frequency of occurrence of U.S. strawberries, this review highlights the acutatum species complex.

In the strawberry industry, strawberry transplants are grown mainly by using starter plant material like runner tips that come from mother plants located in northern latitudes, such as Canada, or mother plants that are located in higher altitudes in the U.S., such as in the mountain regions of North Carolina and California, which have relatively cooler temperatures. Low temperatures may slow disease progression but do not eliminate the pathogen from strawberry transplants [10]. These tips are then rooted in propagation houses in July through September to generate strawberry plug plants at various locations in the U.S. Pathogens such as those under the *C. acutatum* species complex may be transferred to freshly formed plants since transplants are propagated vegetatively [11]. The warm temperatures and humidity in the propagation house create a favorable environment for the moderate-to-high temperature-loving pathogen to thrive. When the infected transplants arrive in fruiting fields, plants are still subjected to ideal environmental conditions for anthracnose disease development, such as extended wetness periods and temperatures between 20 and 30 °C, as overhead irrigation is commonly practiced to aid in the establishment of transplants [12]. Therefore, new technology for detecting the acutatum species complex in asymptomatic nursery plants is a critical need in the North American Strawberry Nursery System to minimize the potential of selling transplants latently infected with *Colletotrichum* spp. to fruit growers [13].

## 2. Anthracnose Diseases in Strawberry

Anthracnose, described as a disease that shows as black, sunken lesions on stems, runners, or fruit and is caused by fungi that generate asexual spores in acervuli, is derived from the Greek roots “anthrak-” (coal) and “-nosos” (disease). Anthracnose fruit rot (AFR) disease on strawberries is caused by multiple strains belonging to the acutatum species complex, which includes 44 species of *Colletotrichum* [14,15,16,17,18]. These diseases are among the most dangerous, causing significant crop losses of up to 70% in commercial production fields planted with susceptible strawberry cultivars [12,19]. *Colletotrichum* spp. has a wide host range and causes several disease symptoms in strawberry plants, including root, fruit, and crown rot, lesions on petioles and runners, and irregular spots on the leaf [20]. *C. acutatum* can infect all parts of the strawberry plant (leaves, petioles, flowers, crowns, and roots) along with the fruit, and lesions may expand and entirely cover the surface of the fruit under favorable conditions (high temperature and humidity), especially on susceptible varieties [21,22].

## 3. Taxonomy of *C. acutatum*

The latest classification of species within the genus *Colletotrichum* is based on their extensive molecular differences, their extensive host ranges, and their diverse lifestyles. The current classification system for the genus *Colletotrichum* consists of more than 280 species, including 16 species complexes (including the acutatum species complex) and 15 singleton species [18]. The fungus *Colletotrichum acutatum* J.H. Simmonds taxonomically belongs to: Fungi, Ascomycota, Pezizomycotina, Sordariomycetes, Hypocreomycetidae, Glomerellales, Glomerellaceae, *Colletotrichum acutatum*, as reported by Simmonds (1965), who described it as a distinct species in Queensland, Australia, in a pathogen survey of fruit rot [23]. The pathogen was previously identified as a species of *Cladosporium* that affected strawberries in Australia, causing mature fruit rot and lesions on the stolon, petiole, and peduncle [24]. In this review, a neighbor-joining phylogenetic tree was generated using the nuclear ribosomal internal transcribed spacer (ITS) region that was retrieved from GenBank of the most updated lists of *Colletotrichum* species that were accepted as members of the acutatum species complex (Figure 1) [18,25]. The majority of the species within the acutatum species complex are recognized as destructive plant pathogens on a global scale [18,26]. The acutatum species complex has been linked to 171 plant species, which are distributed across 129 genera. The majority of these plant species (90.9%) are dicots, while only a small proportion of them are monocots and gymnosperms, accounting for 5.3% and 1.6% of the total, respectively [27]. The acutatum species complex has seven species that were sorted as strawberry pathogens, including *C. acutatum* s.s., *C. fioriniae*, *C. godetiae*, *C. miaoliense*, *C. nymphaeae*, *C. salicis*, and *C. simmondsii* [28].

## 4. Epidemiology and Lifestyle of *C. acutatum*

*Colletotrichum acutatum* inoculum in annual strawberry fields is derived from symptomless infected transplants from nurseries or plug plant production facilities where overhead irrigation might have resulted in conidial dissemination [29]. Under warm temperatures and high humidity, this pathogen rapidly produces conidia, which can spread to flowers and fruit by rain/overhead irrigation, splashing water, and harvesting operations, and consequently show the symptoms of the infections caused by the pathogen, mostly at the fruiting stage. Several studies showed that conidia dispersal occurs within a 25 cm radius of the inoculum source and may vary depending on rainfall intensity and ground cover [30]. This fungus is considered to be hemibiotrophic, where the fungus initially enters a biotrophic phase and then switches to a necrotrophic phase [31]. The pathogen penetrates the cuticle via a specialized cell called an appressorium and grows within the cuticle and cell walls of epidermal, subepidermal, and subtending cells. Then, the fungus produces acervuli as a stroma immediately under the outer periclinal epidermal walls when the cortical tissue is substantially disturbed, and the conidia are released from these acervuli. Although *C. acutatum* may produce quiescent infections on strawberry plants, according to several studies, the epidemiological significance of latent infection before the colonization of fruits and senescent foliage has not been investigated [32].

The quiescent stage (the latent period) is considered to be the time between the fungal infection of the host and the first symptom appearance [33]. The latent period depends on temperature and is between 2–3 days at 25 °C and 6–17 days at 5 °C [34]. The germination of *C. acutatum* conidia, the development of appressoria, latency on non-targeted vegetative organs, including leaves, and serving as a source of overwintering inoculum (Figure 2). In comparison to *C. theobromicola* or *C. gloeosporioides*, *C. acutatum* can produce more conidia at lower temperatures and show the shortest latent period among the three species at 5 and 10 °C. The concentration of inoculum present as latent infections on strawberry plants determines the beginning of disease symptoms. Knowing the minimum inoculum concentration for the start of a disease is informative, particularly in the case of strawberry AFR, as the disease begins as infected transplants from nurseries in commercial fields [35]. Therefore, the development of new tools for the detection of the latent infection at the transplant stage in nurseries is a key factor in controlling AFR diseases on strawberries in fruiting fields.

The primary source of inoculum for fruit infections may be *C. acutatum* appressoria and secondary conidia, which are formed on symptomless leaves and help keep inoculum available throughout the growing season. Wetness for more than 4 h is required for secondary conidia production and appressoria formation [31]. In general, *Colletotrichum* spp. conidia are transmitted from plant to plant in the field primarily by rain splash. Conidia of *C. acutatum* are disseminated over short distances on low-growing crops such as strawberries, and using straw as a ground cover can drastically reduce conidia dispersal [37,38]. Further, the conidia of the three *Colletotrichum* spp. were evaluated with water splash, and it was discovered that the conidia of *C. theobromicola* dispersed over the shortest distance while those of *C. acutatum* spread over the longest distance due to the high production of spores [39]. Although *C. acutatum* can survive in soil and on inanimate surfaces for various lengths of time, depending on the conditions, it appears to compete poorly as a saprophyte [40]. The fungus can survive in the soil for at least two winters with temperatures below 0 °C, causing anthracnose to grow in the following years [41]. Therefore, the application of appropriate disease treatment is required not just for the current year’s crop but also for subsequent seasons. Additionally, it has been proven that weeds host the strawberry infection caused by *C. acutatum*. Nevertheless, a multicrop study conducted in Florida found that *C. acutatum* isolates can be host-specific and offer little threat to other crops [42].

## 5. Detection: Morphological, Molecular, and Remote Sensing

Transplants infected with *Colletotrichum* spp. can spread pathogens from the nursery to the field, and the best management option to avoid the disease is to begin with disease-free planting materials [43]. However, there is no reliable diagnostic method-based protocol to detect latent infection of anthracnose diseases in the early stages of the production cycle in a large production area. Currently, the production of disease-free transplants in nurseries relies on scouting for symptomatic plants and the identification of diseases using colony and spore morphology, which is time-consuming, error-prone, and sometimes inaccurate [41,44]. Molecular techniques such as polymerase chain reaction (PCR), real-time PCR, and ELISA using DNA have become a robust detection and diagnostic tool for plant pathogens [45,46]. However, the polymerase chain reaction (PCR) has some challenges, such as the presence of PCR inhibitors in plant tissues, failure to amplify low DNA concentrations, and the detection of dead pathogens, which can give false positives. Further, all molecular techniques for the diagnosis of plant pathogens are costly, time-consuming, and require the use of highly skilled human resources [47].

Visual inspection, along with field sampling of plant material, is the conventional way of detecting infections in the field, but it is tedious and requires specialized skills [47,48]. The traditional methods cannot detect the latent infection in the early stages of infection. Other laboratory analyses, such as microscopy, molecular, biochemical, and microbiological methods, have been applied for the detection of crop diseases; however, these techniques have disadvantages, as the sampling process is destructive and offers limited diagnostic points, and it is not field-scalable or may not represent accurate field variability [47,49]. Therefore, precise, high-throughput, non-invasive, and field-scalable approaches are required [47,50]. As an alternative in recent years, non-destructive methods such as spectral vegetation indexing, multispectral imaging (MSI), or hyperspectral imaging (HSI) based on ground, aerial, and satellite platforms have emerged that are capable of crop disease diagnostics at high accuracy and on high spatial scales (from leaf to plant to field). Additionally, MSI and HSI could also offer early detection, even before the visual symptoms develop. Such detection can help with proactive management of anthracnose, thereby improving productivity [51].

### 5.1. Remote Sensing of Anthracnose

Remote sensing (RS) with MSI and HSI systems has demonstrated its ability for spatiotemporal vegetation monitoring, including the detection of crop diseases in the early stages [52]. During the infection process to cause disease symptoms, MSI captures light reflected from the surface of the target object, such as a leaf, which is dependent on the changes in both physiological and biological status due to the infection process and subsequent growth stages, e.g., alternations in plant pigmentations such as chlorophyll and carotenoids [51]. Since anthracnose causes physiological, morphological, and plant pigmentation variations, the MSI and HSI techniques could be useful in estimating its incidence. [51]. Remote sensing, such as HSI, was used to detect anthracnose diseases on tea plants with a detection accuracy of 98% for identifying the disease at the leaf level and 94% at the pixel level, where they identified disease-sensitive bands at 542, 686, and 754 nm, which were used to create two disease indices, including the Tea Anthracnose Ratio Index (TARI) and the Tea Anthracnose Normalized Index (TANI) [53].

MSI imaging corresponds to imaging within 3–10 bands of the electromagnetic spectrum in the optical range of 350–1000 nm. Each pixel in the image is represented by a vector, referred to as the spectral signature or fingerprint region of the spectrum [51]. Each fingerprint region of the spectrum has complex absorption sequences due to various bending vibrations within molecules of the plant tissue, and a slight alteration in a compound’s molecular structure will result in a significant change in spectral absorption [54]. The spectral data are helpful, although it may be redundant for adjacent wavelengths. To minimize data size and enhance data utilization efficiency, only significant wavelengths with essential information should be selected for the application of MSI to reduce expenses and increase the speed of plant disease detection [49]. In the field, the sheath blight (ShB) disease on rice, caused by *Rhizoctonia solani*, has been detected with high efficiency using MSI, where five vegetation indices were then calculated from the multispectral images, including the Normalized Difference Vegetation Index (NDVI), Ration Vegetation Index (RVI), Difference Vegetation Index (DVI), Normalized Difference Water Index (NDWI), and Red Edge (RE) [55]. The MSI technique was used to detect the light leaf spot infection with 92% accuracy on oilseed rape (*Brassica napus*) within 13 days before the beginning of visible symptoms, and they used false color mapping of spectral vegetation indices to quantify disease severity and its distribution within the plant canopy in the field [51]. A non-destructive model for the evaluation of firmness, total soluble solids (TSS) content, and ripeness stage in strawberry fruit was established with 100% accuracy using the MSI [56]. Hyperspectral imaging (HIS) has been used to detect anthracnose on strawberry plants, and the spectra of disease in symptomless and symptomatic sections of leaves vary significantly at wavelengths ranging from 540–570 nm to 750–310 nm in the laboratory [57]. Several studies used ultraviolet fluorescence (440–740 nm), multispectral (green [540 nm], red [660 nm], and near-infrared [800 nm]), RGB, and hyperspectral (900–1700 nm) imaging techniques for the detection of crop diseases such as powdery mildew (Erysiphales) on the grapevine (*Vitis vinifera*) [58]. Six machine learning-assisted techniques were devised utilizing the chosen spectral fingerprint characteristics to enable early detection of anthracnose and gray mold diseases on strawberries through the use of a hyperspectral imaging system. The majority of the classification models demonstrated a high level of accuracy (100%) and consistent performance, successfully identifying asymptomatic fungal infections prior to the manifestation of overt disease symptoms, particularly in the strawberry crop [59]. A study investigated the potential of employing hyperspectral imaging (HSI) in conjunction with spectral features, vegetation indices (VIs), and textural features (TFs) to effectively detect gray mold on strawberry leaves. The integration of these combined features in the detection process significantly enhances the accuracy of recognizing strawberry gray mold, enabling the precise identification of infected leaves during the initial stages of infection [60]. A recent study investigated the potential of integrating hyperspectral technology with machine learning and deep learning techniques for the detection of asymptomatic strawberry anthracnose crown rot (ACR), which is caused by *Colletotrichum gloeosporioides*. The accuracy rates of the model test set for healthy, asymptomatic, and symptomatic samples were 99.1%, 93.5%, and 94.5%, respectively [61].

### 5.2. Unmanned Aerial Vehicle (UAV) Platform

Unmanned aerial vehicles (UAVs), typically referred to as drones, have had extensive applications in the past decade for managing crop production operations. A particularly promising use of UAV is in the monitoring of crop health since it may enhance conventional crop monitoring methods, including visual observation to assist in rapid detection, which in turn can have a significant positive effect on crop yield and quality [62]. UAVs are used with MSI or HSI systems to provide high spatial resolution images at flexible flight schedules and short data-acquisition timeframes [52]. Additionally, the imaging data technology based on UAVs has been implemented effectively in diverse applications, including the rapid evaluation of crop vigor and soil characteristics, crop water requirements, disease infestation, and yield prediction [63]. Three platforms, including UAV, sentinel 2, and planet-scope satellite platforms with multispectral (MS) imagery systems, were evaluated based on the analysis of the spatial resolution using soil-adjusted vegetation index (SAVI) to monitor onion crops in the field, and the best result was achieved with the images provided by the UAV platform, which could give more detailed images at critical moments in the crop cycle [64]. Small unmanned aerial systems (UAS) equipped with high-resolution visible (red, green, and blue [RGB]) and multispectral imaging techniques were used to detect powdery mildew (PM) in apple orchards with 77% accuracy [65]. Using a mobile platform, three algorithms, including Stepwise discriminant analysis (SDA), Fisher discriminant analysis (FDA), and K-nearest neighbor (KNN) methods with 32 spectral vegetation indices, were applied to train the model to detect anthracnose diseases at different infection stages on strawberries in both indoor and field trails, and the three models’ classification accuracies were 71.3%, 70.5%, and 73.6%, respectively [48]. To our knowledge, there has been no research on the detection of latent infection of anthracnose diseases on strawberries in the field by using MSI technology with UAV. However, this technological breakthrough for the detection of *Colletotrichum* quiescent infection in strawberries is very promising.

## 6. Management: Chemical, Resistance Breeding, Biological and Biorational

Integrated Pest Management (IPM) programs can help producers by combining techniques that focus on long-term disease and pest management, including using pesticides when required, excellent sanitation practices, planting disease-free plants, and modified cultural practices [66]. Several cultural control methods were used to reduce the *C. acutatum* infection, such as the removal of runners from mother plants. Another cultural practice is the reduction of leaf wetness hours by using drip irrigation instead of overhead irrigation, which helps reduce the movement of the conidia from one plant to another through water splash. Further, hot water therapy has been used for a long time to eradicate pest and disease issues, including cyclamen mites (*Phytonemus pallidus* ssp. *fragariae* Zimmerman) and endoparasite nematodes, in dormant strawberry stock. Runner cuttings were collected from mother plants that had been inoculated with *C. acutatum*, and they were immersed for 7 min at 35 °C, followed by 2 or 3 min at 50 °C. Both treatments of cuttings successfully reduced *C. acutatum* infections from over 80% in the controls to between 6% and 17%, respectively [67].

Chemical methods, such as the usage of fungicides, are a common method among growers to control AFR and other strawberry diseases and frequently rely on a calendar schedule of weekly application [68]. Early in the season, between November and December, inoculum levels are low, and typically the environment is unfavorable for *C. acutatum*; therefore, infected plants do not exhibit symptoms. During this time, the first step in chemical management includes the use of low-label rates of broad-spectrum protectant fungicides like Captan. Then, the inoculum levels increase, and the environment will reach the ideal condition for AFR development from January to March; therefore, higher label rates of broad-spectrum fungicides must be applied weekly depending on the detection of latent infections [68]. In the past, mancozeb, carbendazim, prochloraz, and Tecto 60 have all been used as synthetic fungicides to control the *Colletotrichum* spp. that causes anthracnose in fruits, including strawberries [69,70]. In the last update of the 2023 southeast regional strawberry integrated pest management guide focused on plasticulture production, many fungicides are labeled as “excellent” for controlling the AFR of strawberry plants, such as Merivon (fluxapyroxad + pyraclostrobin), Pristine (pyraclostrobin + boscalid), Luna Sensation (fluopyram + trifloxystrobin), Quadris Top (azoxystrobin + difenoconazole), Quilt Xcel (azoxystrobin + propiconazole), Cabrio (pyraclostrobin), Abound (azoxystrobin), Flint Extra (trifloxystrobin), and Miravis Prime (pydiflumetofen + fludioxonil). On the other hand, other common fungicides such as captan and thiram were rated “good” and "fair,” respectively [71]. However, to avoid the development of resistance strains of *C. acutatum*, all fungicides must be used carefully and in rotation, using different active chemical ingredients following the fungicide resistance action group (FRAC) guide.

The anthracnoseresistance in strawberry cultivars has long been a focus in the scientific community around the world, as the use of resistant cultivars is an excellent management technique against anthracnose diseases. However, growers still plant highly susceptible strawberry cultivars such as “Chandler” and “Camarosa” due to their high fruit quality features, despite suffering significant yield losses due to anthracnose diseases [13,72]. The anthracnose-resistance screening procedure implemented by the United States Department of Agriculture (USDA) has proven to be useful in identifying genotypes that are resistant to the disease in seedling progenies derived from the breeding program of North Carolina State University. Notably, a total of over 32,000 strawberry seedlings that exhibited resistance to anthracnose were identified between the years 1998 and 1999 [73]. Field trials were conducted to evaluate several strawberry cultivars against AFR, and high resistance levels were observed on “Sweet Charlie”, “Ruby Gem”, “Florida Elyana”, and “Florida Radiance” cultivars; intermediate susceptibility was observed in “Strawberry Festival” and advanced selection 99–117; conversely, “Albion”, “Camarosa”, “Camino Real”, “Ventana”, “Candonga”, and “Treasure” were evaluated as susceptible or highly susceptible [22]. Recently, commercial strawberry cultivars including “Flavorfest”, “Florida Belle”, “Florida Elyana”, “Pelican”, “Prado”, “Sweet Sensation”, and “Winter Dawn” were labeled as resistant cultivars in the 2023 Southeast Regional Strawberry Integrated Pest Management Guide [71]. In addition, “Dover”, “Florida Radiance”, and “Winterstar” were graded as medium resistance, and some commercial cultivars such as “Carmine”, “Florida Brilliance”, “Ovation”, “Ruby Gem”, and “Sweet Charlie” were evaluated as medium resistance [71].

Strawberry breeders and plant pathologists are currently engaged in the development of strawberry germplasm that is resistant to AFR and ACR diseases [74,75,76,77]. A recent study demonstrated that the ectopic expression of FvChi-14 in *Arabidopsis thaliana* conferred enhanced resistance against *Colletotrichum higginsianum* by regulating the expression of key genes involved in the salicylic acid (SA) and jasmonic acid (JA) signaling pathways, namely AtPR1, AtICS1, AtPDF1.2, and AtLOX3, that could introduce fresh prospects for the development of disease-resistant strawberry cultivars [78]. Further, another study indicated that the transgenic octoploid strawberries that exhibited overexpression of FaMBL1 demonstrated a reduced susceptibility to the fungal diseases anthracnose and grey mold [77]. However, there are several key factors, such as *Colletotrichum* species, isolate, or race, inoculation method, resistance evaluation method, plant tissue evaluation, and environmental conditions, that can influence the susceptibility or resistance of strawberry genotypes to *Colletotrichum* species during evaluation steps [79].

Many biological control methods and biofungicides have been evaluated on fruit crops against *Colletotrichum* spp., but none have consistently demonstrated field efficacy. Several bacterial biocontrol agents, such as *Bacillus* spp., were summarized and evaluated against *Colletotrichum* spp., such as *C. acutatum*, *C. gloeosporioides*, and *C. truncatum* clades, and they exhibit some inhibitory activities due to the production of antifungal activity via secretion of antifungal metabolites and enzymes or induction of disease resistance in fruits under low disease pressure [80]. *B. subtilis*, *P. polymyxa*, and *B. amyloliquefaciens* have generally been the best agents for managing *C. acutatum*. According to reports, *Paenibacillus polymyxa* secretes antifungal enzymes that can break down chitin, amylase, cellulose, and proteins [81]. The conidial germination of *C. acutatum* was decreased by more than 60% by using *Bacillus* spp., which was isolated from the apple phylloplane, due to the production of fixed and volatile compounds [82]. Prestop (*Gliocladium catenulatum*) and PlantShield (*Trichoderma harzianum*), two commercial fungal biocontrol agents, significantly decreased anthracnose development by up to 45% when sprayed three times onto plants between blooming and fruit ripening [83]. Six isolates of yeast (*Saccharomyces cerevisiae*) successfully controlled *C. acutatum* on citrus plants during preharvest due to several actions, including the production of antifungal compounds, competition for nutrients, inhibition of pathogen germination, and the production of killer activity and hydrolytic enzymes when in contact with the fungus wall [84].

## 7. Anthracnose Diseases Management Challenges

Commercial strawberry growers depend on several management strategies, such as disease-free plants, proper irrigation, mulching, good sanitation practices, pesticides, crop rotation, and disease-resistant cultivars, but none of these have achieved effective control. To control different phytopathogenic fungi, including *Colletotrichum* spp., growers rely on the use of expensive fungicide input in strawberry production systems in the Northeast and Mid-Atlantic; however, the usage of agrochemicals in the management of Anthracnose fruit rot (AFR) and crown rot (ACR) diseases in the strawberry system faces many challenges, including (i) The fumigant methyl bromide (MeBr), which has been banned since 2005 in many countries, including the United States, because of its ozone-depleting properties. (ii) In some cases, fungicide applications failed to control anthracnose disease epidemics due to several reasons, including fungicide-resistant fungi [10]. (iii) In the fresh strawberry fruit market, pesticide usage is less desirable to consumers, and disease-free transplants in the field are a good starting point to achieve that [85]. (iv) A few fungicides were effective against diseases caused by *Colletotrichum* spp. on strawberry plants due to the variability of fungicide sensitivity [86]. In addition, AFR is difficult to control since there is no effective protocol to detect non-symptomatic but infected plants to discard those or take appropriate measures, and the pathogen may build up to large levels in the field without being detected, creating the perfect environment for severe epidemics on ripening fruit under disease-favorable weather conditions. All these considerations highlight the importance of viable biologically-based options in strawberry production systems for the management of soilborne diseases and pests, including anthracnose, that can sometimes affect the crowns and roots of strawberries.

## 8. Alternative and Sustainable Integrated Pest Management Strategies for Soilborne Diseases

Sustainable integrated pest management strategies are needed to meet the global demand for food production for an ever-increasing population [87]. Many alternative soil fumigation methods with synthetic chemicals, such as glucosinolate containing *Brassica* spp., are known to release volatile isothiocyanates (ITCs), which are lethal to different soilborne plant pathogens [88]. Several studies have reported that bio-fumigation with ITC-producing plants is effective against some soilborne plant pathogens, including *Rhizoctonia*, *Verticillium*, *Fusarium*, *Pythium*, and *Phytophthora* spp. [89]. However, this fumigation method is not consistent due to the variable concentration of ITCs in different mustard varieties. From the grower’s perspective, the efficacy of biofumigation was investigated on different plants, and the level of adoption was low. The low efficacy of the treatment has been attributed to many factors, including variations in soil texture, moisture, temperature, soil microbial community, and pH [90].

### 8.1. Overview of Anaerobic Soil Disinfestation (ASD)

Anaerobic soil disinfestation (ASD), a preplant soil disinfestation strategy, is another soil bio-rational method that shows promise to control a wide range of soilborne pathogens and plant-parasitic nematodes. This strategy has not been well experimented with in the Northeast U.S. [91,92]. The ASD process depends on adding carbon (C) sources to stimulate microbially driven anaerobic soil conditions in moist soils covered with polyethylene mulch, which is supposed to convert organic material into other organic compounds that should be lethal to soilborne pathogens [93]. This technique causes changes in soil physical and chemical characteristics, such as the formation of volatile fatty acids, a decrease in soil pH, a rise in soil moisture, and changes in soil nutrients as a result of organic matter addition [94]. Another mechanism of ASD against soilborne pathogens is lowering the redox potential below the critical redox potential (about +200 mV), which can reduce the survival stage of soilborne pathogens [95]. In soils treated with ASD, the inoculum of soilborne pathogens such as *V. dahliae* was reduced by 80–100% compared with the nontreated control and produced marketable fruit yields equivalent to fumigation [96]. In addition, root rot disease complexes on tomato plants, caused by some pathogens including *Colletotrichum* spp., *Verticillium dahliae*, and *Meloidogyne* spp., were significantly reduced in ASD-treated soils in high tunnels compared with plants grown in control soils [97]. In a recent study, the application of 20 t/ha of rice bran to ASD-treated soils resulted in a significant decrease in disease severity after 25 and 60 days of incubation in the strawberry production system. However, the application of a 13.5 t/ha dose did not yield the same reduction in disease severity [98]. ASD treatments with brewer’s spent grain (BSG) and carbon sources considerably reduced the seed viability of all weed species and the *Pythium irregulare* inoculum in a greenhouse trial [99]. The present study has revealed that the application of organic materials at varying dose rates has resulted in significant efficacy against soil-borne pathogens, namely *Fusarium* spp. and *Phytophthora* spp., with a range of 69–99% and 63–98%, respectively. Furthermore, the implementation of ASD has led to a notable increase in the levels of soil organic matter and ammonium nitrogen [100]. However, this technique needs to be optimized in terms of engaging beneficial microorganisms to control different soilborne pathogens and enhance plant vigor and productivity at the same time [101]. The mechanisms of ASD are not fully understood; they may be due to the toxic by-products of anaerobic decomposition, volatile compounds, biocontrol by anaerobic soil microorganisms, or oxygen deficiency [91].

### 8.2. Optimizing Anaerobic Soil Disinfestation (ASD) with Endophytic Bacteria

In general, several factors influence the efficacy of ASD, such as the carbon source, the addition of beneficial microorganisms, and environmental conditions including soil types, pH, and temperature. Based on our previous work, BSG was used effectively as a carbon source to support soil microbial growth in ASD applications in field trials [102]. Engaging beneficial microorganisms such as endophytic bacteria that are used as biofertilizers or biostimulants with the ASD technique could generate a powerful tool to control soilborne pathogens and improve the growth and yield of strawberries, which may play a crucial role in sustainable crop production in the future [103]. Beneficial microorganisms can improve plant nutrition, support plant development under natural or stressed conditions, and increase the yield and quality of many important crops [104]. In the interaction between beneficial microorganisms and plants, these organisms act as nutrient suppliers, phytohormone producers, plant growth enhancers, biocontrol agents of phytopathogens, and improvers of soil structure [105,106]. Root dipping of seedlings (plug plants), followed by spray treatments of both probiotic bacteria, including *Bacillus amylolequefaciens* (BChi1) and *Paraburkholderia fungorum* (BRRh-4) on leaves in the field, dramatically enhanced the fruit yield of strawberry plants by 48% compared to the untreated controls [107]. In the greenhouse, three strains of *Bacillus velezensis,* an endophyte bacterium, significantly suppressed strawberry pathogen growth (*C. gloeosporioides*) and increased marketable fruit yields in the field [108]. The gray mold disease in strawberry plants, caused by *Botrytis cinerea*, was controlled by five different isolates of *Bacillus* spp. via the production of diffusible and volatile antifungal chemicals [109]. The severity of Rhizoctonia root rot disease on Viburnum plants (*Viburnum odoratissimum*) was reduced on both greenhouse and field trials using the TerraGrow product, which is a complex of five *Bacillus* strains, including *B. licheniformis*, *B. subtilis*, *B. pumilus*, *B. amylolquefiens*, and *B. megaterium* [110]. In a perennial strawberry production system, the combination of beneficial microbes and ASD enhanced plant vigor and fruit yield and suppressed the weed population and pathogenic microbes compared with untreated plants [111].

## 9. Future Perspectives

Strawberry production has increased around the world in the past few years due to rising demand. Cutting-edge research programs are ongoing to solve problems that threaten strawberry production while also enhancing fruit quality to meet customer demands. There is an urgent need for the adoption of sustainable alternative disease management measures that pose little threat to human health and the ecological system [111]. Integrated Pest Management (IPM) systems, which combine biological, cultural, and chemical tools with other supporting technologies, are effective, efficient, and sustainable. The combination of ASD with beneficial microbes is being introduced lately in agricultural practices in place of fungicides and soil fumigants due to their economic viability and environmental friendliness [111]. It appears that ASD with different C sources is a viable approach to controlling diseases, increasing yield, and improving soil, especially in limited sources, organic farming, and smallholder farming. Furthermore, integrating ASD with beneficial microbes could reduce the initial investment for ASD treatments alone and create a powerful tool for pest management, including fungi, bacteria, nematodes, and weeds in the strawberry production system. Future research should concentrate on understanding how to incorporate suitable beneficial microbes to control specific pathogens, as well as better understanding which mechanism(s) are responsible for disease control in different situations.

A precise detection method for pathogens is a requirement for the application of the appropriate disease control techniques. The current review may also highlight the need for a rapid, non-destructive, and accurate method for anthracnose fruit rot (AFR) at the early stage of the infection (latent period). Further, strawberry growers benefit from early incubation stage identification because it allows them to immediately remove contaminated plants before the disease spreads and causes further damage. In recent years, a combination of small unmanned aerial systems (UAS) equipped with multispectral imaging (MSI) sensors, which integrate spectral and image data, has demonstrated considerable benefits for non-destructive inspections, plant disease identification, and the safety of agricultural products. We believe this review of using remote sensing to diagnose anthracnose fruit rot (AFR) will provide novel thoughts and encourage the development of appropriate theories, methods, and tools to monitor strawberry transplants in nurseries, which are considered the main source of inoculum for the production farms.

## Figures and Tables

**Figure 1 plants-12-03704-f001:**
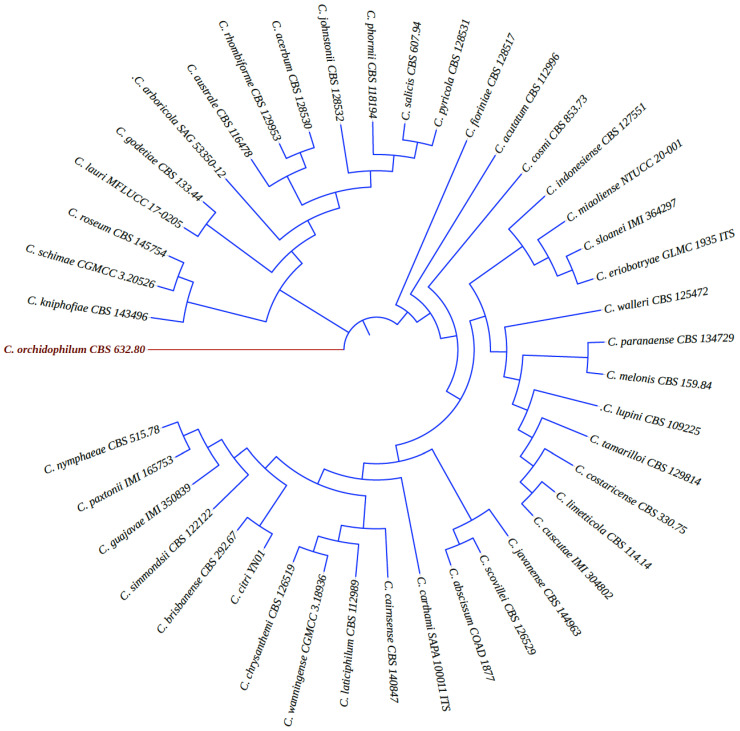
Phylogenetic analysis was generated using a neighbor-joining tree based on 42 strains of *Colletotrichum* spp. that were accepted as members of the acutatum species complex [18,25]. The analysis of sequences of internal transcribed spacers (ITS) were retrieved from GenBank. *Colletotrichum orchidophilum* was employed as an outgroup strain and is highlighted in red color.

**Figure 2 plants-12-03704-f002:**
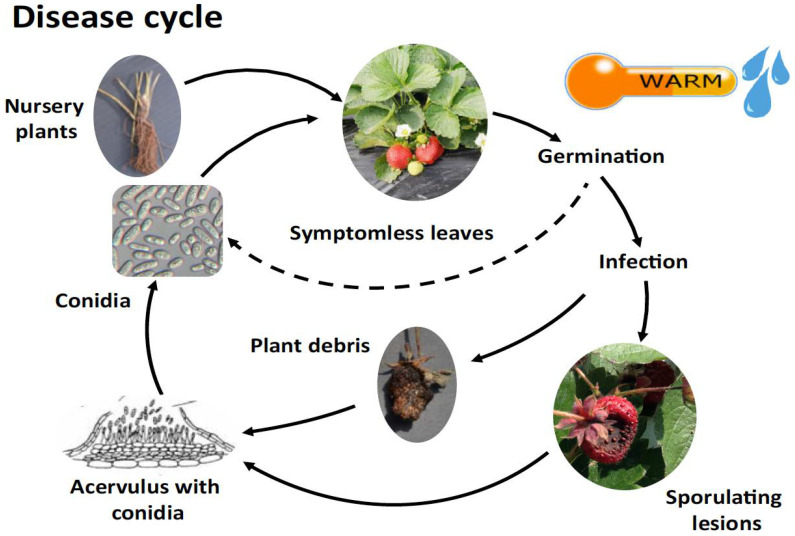
The disease cycle of anthracnose fruit rot caused by (AFR) *C. acutatum* [36].

## Data Availability

Not applicable.

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
