# Peer review of "New Insights in the Detection and Management of Anthracnose Diseases in Strawberries"

_plants, 2023, doi:10.3390/plants12213704_

Round 1
Reviewer 1 Report
Comments and Suggestions for Authors
Well written manuscript.
One critcal comment: the authors should discuss the way of resistance breeding for a sustainable planting of strawberrries. Is there known anything about a geneitc based tolerance/resistance?
Reviewer 2 Report
Comments and Suggestions for Authors
Dear Authors,
Please find below and attached my comments and suggestions for your work.
Good luck!
Kind regards,
The Reviewer
Review Report Form
Journal: Plants (ISSN 2223-7747)
Manuscript ID: plants-2561679
Type: Review
Title: New Insights in the Detection and Management of Major Foliar and Root Diseases of Strawberries
Authors: Mahfuz Rahman * , Jayesh Samtani , Aljawasim Baker
Section: Plant Protection and Biotic Interactions
Special Issue: Interactions between Colletotrichum Species and Plants III
Submission Date: 31 July 2023
Dear Authors,
My comments regarding your manuscript (Review) are below. Thank you for the opportunity to read this study!
(A.) General overview of the article and strong points:
Ø First of all, I would like to mention that I highly enjoyed reading and analyzing your study: the Review manuscript entitled “New Insights in the Detection and Management of Major Foliar and Root Diseases of Strawberries”. I believe that this study is highly representative and has the capacity to rigorously synthetize the aspects intended.
Ø Second of all, a particular attention was shown by the authors to all the aspects surrounding these new insights in the detection and management of major foliar and root diseases of strawberries, as well as to the future perspectives regarding these methods. This might represent a valuable starting point for new researches performed by the authors themselves or by other researchers in the field.
Ø General background of the study and aim of the study: The authors highlighted that the Anthracnose and root rot are major foliar and root diseases of strawberry, respectively that cause most yield and quality losses. The authors mentioned that Anthracnose is caused by multiple species belonging to Colletotrichum spp complex. Also, the authors pointed out that this disease alone can cause up to 70% yield loss in North America. Moreover, the authors highlighted that Colletotrichum spp. cause several disease symptoms on strawberries, including root, fruit, and crown rot, lesions on petioles and runners, and irregular black spots on the leaf. In the same line with the aspects mentioned above, the authors pointed out that, in many cases, a lower level of infection on foliage remains non-symptomatic (quiescent), posing a challenge to growers as these plants can be a significant source of inoculum for the fruiting field.
Ø Methodology of the study: In terms of the methods and methodology, the authors stressed that the reliable detection methods for quiescent infection should play an important role in preventing infected plants' entry into the production system or guiding growers to take appropriate preventative measures to control the disease. Also, the authors stressed the fact that the selection of highly effective fungicides and including them in a schedule to prevent the development of fungicide resistance in fungal populations will remain an area of continued research.
Ø Results of the study: In terms of the results of this current study, it ought to be mentioned that that the authors have emphasized that the disease management methods should entail sensitive molecular testing of suspected fungal isolates for resistance, rotation of products, and monitoring of quiescent infections in transplant materials. Moreover, the authors emphasized the fact that testing and the inclusion of non-chemical methods such as biologicals and biorational treatments especially for soilborne pathogens will also be necessary to make disease management economically feasible and sustainable.
(B) Suggestions meant to improve your current manuscript:
Distinguished Authors I would kindly like to suggest the following aspects that are meant to improve your study – Review:
(1.) There are some English language improvements and slight corrections that need to be made. So, my recommendation is to carefully proofread the entire manuscript.
(2.) Since I closely analyzed the format of the article, I noticed that, according to the guidelines of the publisher, some adjustments need to be made. It would also be highly recommendable to include in the abstract of your study more highly relevant details that refer to the research objectives and the methodology used. This would definitely be considered a plus for your scientific work.
(3.) A great idea would be inserting in your article a few ideas concerning the correlation between sustainability assessment, the effects of the COVID-19 pandemic and the COVID-19 global crisis, sustainability, and Sustainable Development Goals, while concentrating on sustainable management techniques, and the new insights in the detection and management of major foliar and root diseases of strawberries, since these are main objectives today. There are a few interesting studies that I read lately and that I can mention: Measuring Progress Towards the Sustainable Development Goals: Creativity, Intellectual Capital, and Innovation. In C. Popescu (Ed.), Handbook of Research on Novel Practices and Current Successes in Achieving the Sustainable Development Goals (pp. 125-136). IGI Global. https://doi.org/10.4018/978-1-7998-8426-2.ch006; OECD. 2022. Beyond the bottom line: Measuring the non-financial performance of firms through the lens of the OECD Well-being Framework. https://oecdstatistics.blog/2022/12/12/beyond-the-bottom-line-measuring-the-non-financial-performance-of-firms-through-the-lens-of-the-oecd-well-being-framework/; OECD Agriculture and Food Policy Reviews. 2023. ISSN: 27102610 (online). https://doi.org/10.1787/f061e50b-en.
Dear Authors, thank you, once again, for the chance to read this study! Good luck!
Kind regards,
The Reviewer

Minor editing of English language required.
Reviewer 3 Report
Comments and Suggestions for Authors
1). Manuscript ID: Plants-2561679
2). Manuscript Title: New Insights in the Detection and Management of Major Foliar and Root Diseases of Strawberries
3). General Comments
--Please follow the Journal format while revising the manuscript.
--Add scientific authority at the end of binomial names of all species when they are mentioned for the first time in the manuscript.
--Include full forms of all abbreviations/acronyms mentioned in the manuscript.
--Minor English changes required throughout the manuscript. Please check the manuscript carefully before resubmission.
E.g., Page 3, Line 1: Add space between “symptoms” and “in.”
Comments on the Quality of English Language--Minor English changes required throughout the manuscript. Please check the manuscript carefully before resubmission.
E.g., Page 3, Line 1: Add space between “symptoms” and “in.”
